# A Psychosocial Intervention’s Impact on Quality of Life in AYAs with Cancer: A Post Hoc Analysis from the Promoting Resilience in Stress Management (PRISM) Randomized Controlled Trial

**DOI:** 10.3390/children6110124

**Published:** 2019-11-02

**Authors:** Angela Steineck, Miranda C. Bradford, Nancy Lau, Samantha Scott, Joyce P. Yi-Frazier, Abby R. Rosenberg

**Affiliations:** 1Center for Cancer and Blood Disorders Center, Seattle Children’s Hospital, Seattle, WA 98105, USA; angela.steineck@seattlechildrens.org (A.S.); samantha.scott@seattlechildrens.org (S.S.); joyce.yi-frazier@seattlechildrens.org (J.P.Y.-F.); 2Department of Pediatrics, University of Washington School of Medicine, Seattle, WA 98105, USA; nancy.lau@seattlechildrens.org; 3Children’s Core for Biomedical Statistics, Center for Clinical and Translational Research, Seattle Children’s Research Institute, Seattle, WA 98105, USA; miranda.bradford@seattlechildrens.org; 4Center for Clinical and Translational Research, Seattle Children’s Research Institute, Seattle, WA 98105, USA; 5Cambia Palliative Care Center of Excellence, University of Washington, Seattle, WA 98105, USA

**Keywords:** adolescent and young adult, pediatric oncology, resilience, quality of life

## Abstract

Promoting Resilience in Stress Management (PRISM), a psychosocial intervention for adolescents and young adults (AYAs) with serious illness, enhances resilience resources via four skills-based training sessions. A recent randomized controlled trial showed PRISM improved health-related quality of life (HRQOL) compared to usual care (UC). This post hoc exploratory analysis aimed to better understand the effect of PRISM on HRQOL by describing changes in HRQOL subdomain scores. English-speaking AYAs (12–25 years) with cancer were randomized to PRISM or UC. At enrollment and six months later, HRQOL was assessed using the Pediatric Quality of Life Inventory (PedsQL) Generic Short Form (SF-15) and Cancer Module. Scores at each time point were summarized descriptively and individual HRQOL trajectories were categorized (<70 vs. ≥70). “Positive” trajectories indicate participants maintained scores ≥70 or improved from <70 to ≥70 during the study period. Baseline assessments were completed by 92 participants (48 PRISM, 44 UC); six-month assessments were completed by 74 participants (36 PRISM, 38 UC). For the SF-15, positive trajectories in psychosocial domains were more common with PRISM; trajectories in the physical subdomain were similar across groups. For the Cancer Module, positive trajectories were more common with PRISM in the following subdomains: nausea, treatment anxiety, worry, cognitive, physical appearance, and communication. From this, we conclude PRISM may improve HRQOL, especially in psychosocial domains of wellbeing.

## 1. Introduction

Pediatric patients undergoing cancer-directed therapy are known to have poorer health-related quality of life (HRQOL) compared to their well peers [1,2]. Poor HRQOL correlates with increased rates of depression and anxiety [3,4,5], poor treatment adherence [6,7], and inadequate follow-up care [8,9]. Individuals who survive cancer diagnosed as an adolescent or young adult are at higher risk for psychological distress compared to individuals diagnosed as younger children or older adults [10]. This distress has longstanding consequences, with reported associations between poor psychosocial outcomes and physical limitations [11,12,13,14], difficulties developing intimate relationships [15,16], lower educational attainment [17,18], and increased economic burden among adult survivors of childhood cancer [19,20]. The disruption of normal skill development necessary for navigating challenges unique to this age group, in combination with inadequate psychosocial support, is thought to contribute to these poor outcomes [21].

In order to mitigate the effect of this disruption, a growing number of evidence-based psychosocial interventions have been designed for adolescents and young adults (AYAs) with cancer [22], defined here to be age 12–25 years. Interventions targeting AYAs receiving cancer-directed treatment include Re-Mission, a computer game engineered to reinforce positive self-care behaviors, such as medication adherence and relaxation techniques [23]; a computer-based educational intervention targeting coping skills while providing information about the AYA’s disease, treatment, and potential late effects [24]; and a therapeutic music video intervention for AYAs undergoing hematopoietic stem cell transplant [25]. Neither Re-Mission nor the computer-based educational program were associated with a significant change in patient-reported HRQOL; HRQOL was not directly reported for the therapeutic music video intervention [23,24,25]. Other interventions have focused on cancer survivors rather than those currently receiving treatment [26,27]. Both interventions report suggested HRQOL benefits, although limited by occurring late in the cancer experience [26,27]. Altogether, the relationship between interventions for AYAs with cancer and HRQOL remains poorly understood.

Resilience is the capacity to maintain physical and emotional well-being in the face of stress [28]. It can be strengthened by cultivating skills, such as stress-management, goal-setting, cognitive reframing, and meaning-making. In older adults with cancer, the promotion of “resilience resources” is associated with decreased rates of depression, improved psychosocial health [29], and improved HRQOL [30,31]. Promoting Resilience in Stress Management (PRISM) is a brief, psychosocial intervention that aims to strengthen resilience-building skills in AYAs with serious illness [32,33,34]. In a recent phase II randomized controlled trial (RCT) [33,34], AYAs diagnosed with cancer were randomized to usual care (UC) alone or UC plus PRISM. Primary results demonstrated improved cancer-related HRQOL [33,34]. Although point estimates suggested a similar improvement in generic (non-disease-specific) HRQOL, findings were not statistically significant. To better understand these results, this post hoc exploratory analysis aimed to explore the effect of PRISM on HRQOL by describing individual subdomain scores in PRISM and UC. Given the nature of the intervention, we hypothesized that PRISM’s benefit would be greatest in subdomains representative of psychosocial wellbeing. Ultimately, the goal of this analysis was to generate hypotheses regarding opportunities for intervention refinement and future patient-centered research.

## 2. Materials and Methods

### 2.1. Design, Setting, and Participants:

The PRISM phase II, parallel, 1:1 randomized controlled trial was conducted at Seattle Children’s Hospital (SCH) between January 2015 and October 2016 [33,34]. Eligible participants were 12–25 years old, fluent in English, and either diagnosed with a new cancer (between one and ten weeks prior to enrollment and receiving systemic chemotherapy) or with “advanced cancer” (progressive, recurrent, or refractory disease at any time since initial diagnosis). Individuals with pre-existing or cancer-associated cognitive disabilities were excluded. The study was approved by the SCH Institutional Review Board. Informed consent was obtained from all individual participants included in the study. This trial is registered at ClinicalTrials.gov (NCT02340884).

### 2.2. Recruitment and Randomization:

As described previously, consecutively eligible participants were identified by study staff using clinic rosters and approached regarding study participation [33,34]. All individuals over the age of 18 years provided informed consent for study participation. For study participants under the age of 18 years (12–17 years), parental or guardian informed consent was required for study participation and participant written assent was provided. Participants were enrolled and randomized one-to-one to usual care alone (UC, control) or usual care plus PRISM (intervention) [33,34]. Randomization was completed using a permuted block algorithm until target enrollment (*n* = 100) was achieved [33,34]. Because intervention materials are tailored to teens (12–17 years-old) or young adults (18–25 years-old), the randomization was stratified by age. Blinding of study staff prior to randomization assignment and staff assigned to collect outcome data remained blinded to assignment.

### 2.3. Intervention:

PRISM is a psychosocial intervention, developmentally designed for AYAs experiencing serious chronic illness based on theories of resilience, stress, and coping [32]. It aims to strengthen four key resilience resources: stress-management, goal-setting, cognitive reframing, and meaning-making. The intervention is delivered one-on-one by trained, college-educated interventionists, who receive a minimum of eight hours of protocol-derived training. Four skill-building sessions, 30–60 min in duration, are delivered approximately every other week. Sessions are highly interactive, with the interventionist encouraging participant engagement and application of each targeted skill. The intervention concludes with an optional facilitated family meeting, where the AYA and family members review intervention topics and reinforce skills together. Prior studies have established its feasibility, acceptability, and efficacy and describe intervention implementation in detail [32,33,34]. At Seattle Children’s Hospital, usual care includes a comprehensive psychosocial assessment at the time of diagnosis and upon family request or staff referral thereafter.

### 2.4. Procedures:

As part of the phase II RCT, all participants were invited to complete a comprehensive Patient-Reported Outcome (PRO) assessment including AYA age-validated instruments assessing HRQOL. Surveys were completed at time of enrollment and six-months later. Study staff provided reminders for survey completion prior to each due date and once weekly for up to three weeks for any surveys that were not returned. If not received within twelve weeks of the due date, surveys were considered missing. Participants received a $25 gift card following completion of the baseline PRO survey and a $50 gift card following completion of the six-month PRO survey.

### 2.5. Patient-Reported Outcomes:

HRQOL was evaluated using instruments from the PedsQL library, a battery of valid and reliable PRO instruments used successfully in prior research in AYAs with cancer with low rates of refusal and missing data [35]. The PedsQL 4.0 Generic Core Scale [1,36,37] is a nonspecific PRO instrument developmentally appropriate for children, adolescents, and young adults ranging from age five to twenty-five years. This instrument encompasses subdomains representing the core dimensions of health according to the World Health Organization including physical, emotional, and social wellbeing, as well as “school wellbeing.” We used the abbreviated 15-item instrument (SF-15) instead of the “full” 23-item instrument to minimize survey burden. Compared to the 23-item instrument, the SF-15 omits questions regarding pain, fatigue, difficulty sleeping, ability to bathe independently, and functional ability compared to peers [38]. Psychometric properties of the 15- and 23-item PedsQL generic instruments are comparable [38]. The PedsQL Cancer Module is a 27-item instrument assessing subdomains specifically related to the cancer experience, including pain, nausea, procedural anxiety, treatment anxiety, worry, cognition, perceived physical appearance, and communication [1].

PedsQL items are rated on a five-point Likert scale and scores are transformed to a 0–100-point scale. Higher scores represent better quality of life. Individual subdomain (“Scale Scores”) are the mean of the individual item scores (sum of individual item scores divided by the number of questions answered). A “Total Scale Score” is calculated as the sum of all items on all subdomains divided by the total number of questions answered. Internal consistency of the total score for the adolescent Generic Core Scales for the total sample is 0.92 among healthy adolescents and adolescents with cancer. Individual subscale internal consistency ranges from 0.75 to 0.88 [1,36,38]. For the PedsQL 4.0 Generic Core Scale, the minimal clinically important difference (MCID) is estimated to be 4.4 points for the total score, and 6.6–6.9 points for the subscale scores [39,40].

### 2.6. Study Outcomes:

The primary outcome for the PRISM RCT was the six-month time-point patient report of resilience. Results of the primary outcome analysis have been reported previously [33]. In this post hoc analysis, we report on the secondary outcome of HRQOL, including both cancer-specific and generic HRQOL.

### 2.7. Statistical Analyses:

In order to detect a minimal clinically important difference of 4.7 points for the primary outcome of the study (patient-reported resilience), the study was designed to provide 80% power with a two-sided alpha of 0.05. A sample size of 90 participants was selected based on these parameters. A priori power calculations for the secondary outcome of HRQOL were not completed.

Participant characteristics at enrollment for patients who completed baseline assessments were summarized using counts and percentages. We calculated means and standard deviations to summarize total and subdomain scores for SF-15 and disease-specific quality of life scores at baseline and 6 months. Subjects who completed only baseline assessments were included in the score summaries.

We used an established cut-point of 70 to define HRQOL trajectories [1,39,40,41]. Baseline and six-month scores were categorized as “good” = 70–100 or “poor” = 0–69. Individual patient response trajectories from baseline to follow-up were categorized as: (a) ≥70 at BL, <70 at six months: “deteriorated”, (b) <70 at BL and six months: “consistently at risk”, (c) ≥70 at BL and six months: “consistently well”, (d) <70 at BL, ≥70 at six months: “improved”. Both (c) and (d) were considered to be “positive” HRQOL trajectories. Subjects who completed both baseline and six month assessments were included in the trajectory analysis.

To assess internal consistency of the sample, we calculated Cronbach’s alpha with one-sided (lower) 95% confidence intervals for each subscale and the total scale for both the SF-15 and the Cancer module.

Due to the post hoc exploratory nature of the analysis and small subsample sizes, we did not test statistical significance of intervention effects, but rather chose to summarize the proportion of participants with positive and negative trajectories in each study arm using 95% confidence intervals and graphical representations. Statistical analyses were performed using Stata version 15 (StataCorp., College Station, TX, USA).

## 3. Results

The identification, enrollment, and randomization results of this study have been reported elsewhere [32,33,34]. Enrollment took place over a 22-month period. Briefly, 483 pediatric patients with cancer were screened for eligibility, 130 individuals met eligibility criteria, 100 participants enrolled (76%). Fifty individuals were randomized to each of the study arms. One participant randomized to the PRISM arm was immediately withdrawn following randomization and prior to receiving the baseline survey instrument because it was disclosed that they were not fluent in English. Of the remaining 99 participants, 92 completed baseline data and of these, 44 received UC and 48 received UC plus PRISM (Table 1). Age, race, cancer-type, and presence of advanced cancer were comparable between arms, while the number of females was lower in the intervention (*n* = 16) than UC (*n* = 24).

Most participants were younger than 18 years-old at enrollment (73%) and leukemia/lymphoma was the most common cancer-type (65%). Twenty-six percent of participants had advanced cancer at enrollment. Attrition during the 6-month study period was predominantly due to disease complications or death. Demographic characteristics of those who completed six-month assessments were similar to those in the initial sample. For example, of the 74 participants [36/48 (25% attrition) PRISM; 38/44 (14% attrition) UC] who completed six-month PROs, 72% were 12–17 years old and 23% had advanced cancer at enrollment. Internal consistencies for the SF-15 ranged from 0.71 to 0.91 on the subscales at baseline and 0.67 to 0.92 at follow up (Appendix A). Internal consistencies for the Cancer module ranged from 0.76 to 0.89 on the subscales at baseline and 0.81 to 0.93 at follow up (Appendix A).

### 3.1. Generic Quality of Life

The mean baseline SF-15 Total Score was 59 (SD 21) for UC and 62 (SD 16) for PRISM (Appendix A). Scores at six months were 60 (SD 19) for UC and 67 (SD 15) for PRISM. Compared to UC, participants who received PRISM had a higher proportion of positive long-term global HRQOL trajectories (PRISM = 47%, 95% CI: 32–63 vs. UC = 26%, 95% CI: 15–42) (Table 2, Figure 1). More PRISM recipients than UC recipients improved (PRISM: 33% vs. UC: 0%). The mean change among participants with improved scores was +26 (SD 12) and the minimum change in this group was +4. Similarly, the mean change for the group of participants whose scores declined was −26 (SD 15), with the minimum change −7. Across all three psychosocial subdomains, more participants who received PRISM had positive trajectories (emotional: PRISM = 58% vs. UC = 37%; social: PRISM = 83% vs. UC = 66%; school: PRISM = 45% vs. UC = 35%). The proportion with positive physical subdomain trajectories was similar across study arms (PRISM: 36% vs. UC: 34%).

### 3.2. Cancer-Specific Quality of Life

The mean baseline Cancer Module Total Score was 65 (SD 17) for UC and 66 (SD 16) for PRISM (Appendix A). Mean scores at six months were 64 (SD 20) for UC and 72 (SD 11) for PRISM. Compared to UC, the proportion of participants with positive trajectories was higher for PRISM recipients in the nausea, treatment anxiety, worry, cognitive, physical appearance, and communication subdomains (nausea: PRISM = 64% vs. UC = 39%; treatment anxiety: PRISM = 72% vs. UC = 61%; worry: PRISM = 50% vs. UC = 24%; cognitive: PRISM = 58% vs. UC = 42%, physical appearance: PRISM = 50% vs. UC = 42%; communication: PRISM = 69% vs. UC = 55%) (Table 2, Figure 2). The greatest advantage was observed in nausea, worry, and cognitive domains. At least 50% of PRISM recipients had positive trajectories in seven of the eight subdomains, compared to three out of eight subdomains for UC recipients (Figure 2).

For two domains, the proportion of participants with positive trajectories was lower among PRISM recipients: pain and procedural anxiety (pain: PRISM = 36% vs. UC = 39%; procedural anxiety: PRISM = 58% vs. UC = 74%).

## 4. Discussion

The period immediately following a diagnosis of cancer is stressful and associated with poor HRQOL [42,43]. However, evidence-based interventions for AYAs early after diagnosis and/or during cancer therapy are lacking and the impact on quality of life is not well understood. PRISM works by strengthening pragmatic psychological and coping skills, teaching how to be aware of and reappraise one’s thoughts, plan for what one identifies as important, and identify what one values in life [32,33,34]. A recent clinical trial suggested PRISM has the potential to improve HRQOL in AYAs with cancer. The objective of this post hoc analysis was to better understand the effect of PRISM across subdomains of general and cancer-specific HRQOL.

We expected more positive trajectories with PRISM in subdomains both representing perceived psychosocial wellbeing and relevant to the illness experience unique with a cancer diagnosis. For generic HRQOL, we identified the social and emotional subdomains as most representative. We did not expect a difference in the school subdomain because many AYAs undergoing cancer-directed therapy do not attend school. However, we found more positive trajectories with PRISM in subdomains representing psychosocial wellbeing, including the school subdomain. Individual items from the school domain query cognition and may have been extrapolated to other dimensions of quality of life by respondents, which may have accounted for our findings. For disease-specific HRQOL, we expected more positive trajectories in subdomains representing perceived psychosocial wellbeing that are potentially modifiable by applying reframing or stress management skills. This included procedural anxiety, treatment anxiety, worry, perceived physical appearance, and communication. However, we found the greatest benefit in the nausea, worry, and cognitive subdomains among PRISM recipients.

Altogether, we suspect that deviations between our findings and our expected results may be attributable to either or both of the following: It is possible that the subdomains with the greatest change may be those that query values and perceptions most directly. For example, the worry subdomain prompts reflection on degree of worry associated with how well treatments are working. This is compared to the procedural anxiety subdomain, which inquires whether or not procedures are associated with pain. It is also possible that participants may have applied skills they found most helpful to areas they considered in greatest need of improvement. For example, if a participant recognizes that they worry often and they find mindfulness based stress reduction to be helpful, they may apply this skill to this specific instance more readily. Similarly, it is likely that not all skills are found to be equally valuable to all participants and, if an area of one’s experience is not personally identified to be important to change, it is unlikely that a participant will apply any skills to that area. For example, not all participants may find cognitive reframing equally helpful and so would be less likely to apply it to real-life situations.

There are also important exceptions to the positive trends we observed with PRISM. For example, PRISM recipients had less positive trajectories with respect to pain and procedural anxiety. Notably, no adverse events were reported by study participants in the intervention group or the control group during the study period, including increased reports of pain or anxiety with procedures. Reasons for these exceptions to PRISM’s success include the possibilities that PRISM is insufficiently equipped to address this specific illness experience and/or there are additional and specific needs regarding procedural worries or pain. With no adverse events reported during the study period, we think this is more likely than a true negative impact resulting from the PRISM intervention. PRISM is designed to introduce participants to pragmatic coping skills. If a participant does not identify a certain skill to be valuable to their own experience, that skill will not be applied. Moreover, if a participant does not identify a particular situation as an area in need for improvement, skills will not be applied in that instance. For example, the fact that procedural anxiety did not improve may indicate participants did not identify anxiety with procedures as an area relevant or meaningful for them to apply the skills learned from PRISM. Specific to the domain of pain, the instrument used in this study measures wellbeing associated with pain rather than pain interference. PRISM is unlikely to change whether or not a participant experiences pain. However, it may change the consequences of pain and this was not measured. Interventions such as mindfulness and meditation have been reported to produce unintended negative consequences; however, the mechanism is not well understood [44,45]. This has not been reported specifically with pain control, but may also represent an alternative explanation for our results. Regardless, these areas represent potential opportunities to improve the intervention among subsets of patients and specific attention to adverse events in these two domains will occur in future PRISM intervention studies.

Evaluating the intervention’s impact on HRQOL by subdomain, rather than by total score, adds to our understanding of how the intervention impacts specific elements of participants’ cancer-experiences. For example, the fact that positive communication trajectories were more common in PRISM recipients suggests that PRISM skills may empower AYAs to engage in communication with their families and oncology team about their diagnosis and treatment. Empowering AYAs to be engaged in their own health care not only increases AYA participation in the advancement of AYA-specific oncology research, but is also associated with improved HRQOL [46]. The use of subdomain scores in evaluating an intervention’s effect also aids in the identification of areas of wellbeing that may be most amenable to change. With this knowledge, more precise refinement of the intervention may be achieved. For example, in future iterations of the PRISM intervention, we plan to capitalize on a potential improvement in patient engagement in communication with the addition of a fifth skill-building session for AYAs with advanced cancer (ClinicalTrials.gov identifier: NCT03668223).

By definition, generic PROs are intended to assess an individual’s health status compared to the general population. In contrast, disease-specific measures are meant to assess individuals with a shared disease experience. The use of generic quality of life measurement has been favored as a primary outcome in the literature [47,48]. However, when testing an intervention designed for individuals with a shared chronic disease experience, the use of disease-specific measures may provide a better evaluation of the intervention’s impact. Here, this is particularly evident in our assessment of psychosocial wellbeing. Looking at the generic HRQOL scores, we found, positive trajectories in the psychosocial subdomains (social, emotional, and school wellbeing) were more common with PRISM. Including a disease-specific HRQOL instrument supports our conclusion that PRISM provides a psychosocial wellbeing benefit, with positive trajectories being more common with PRISM for subdomains such as worry, treatment anxiety, and communication. It additionally adds to our understanding by suggesting participants may be applying these skills to cope with aspects unique to the cancer experience, such as nausea and perceived physical appearance. This information may be leveraged in future iterations of the intervention by augmenting certain skills for specific risk groups.

We acknowledge several limitations of this analysis. Most importantly, this was a post hoc analysis of a secondary study outcome and, as such, we lacked power to confirm the statistical significance of our findings with classical hypothesis tests. Instead, the descriptive data may be useful in identifying patterns to guide future research and application of the PRISM intervention. Additional limitations which may limit the generalizability of our results are the small population size, single institutional setting, and including only English-speaking participants. Additionally, as this trial was conducted at a pediatric cancer center, only younger AYAs are represented. Notably, generic HRQOL was measured using the abbreviated PedsQL short form, rather than the longer PedsQL 4.0 generic form. This may have limited our ability to detect a significant difference in generic HRQOL in the primary analysis, and may have led to subtle differences within subdomains for this post hoc analysis.

## 5. Conclusions

This AYA-targeted resilience intervention led to improved generic and cancer-specific HRQOL trajectories. Evaluating HRQOL at the subdomain level and including disease-specific measures of HRQOL enriches our understanding of potential intervention effects. For example, positive communication trajectories were more common in PRISM recipients, suggesting PRISM shows promise in promoting self-efficacy and patient activation in AYAs with cancer. Future research will better characterize risk groups to inform intervention modification and provide the greatest benefit to specific patient populations.

## Figures and Tables

**Figure 1 children-06-00124-f001:**
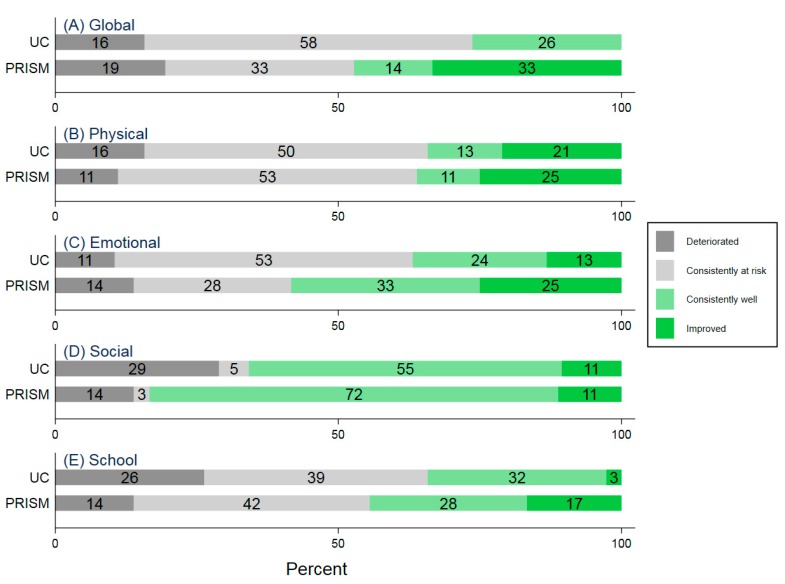
Generic health-related quality of life (HRQOL) trajectories.

**Figure 2 children-06-00124-f002:**
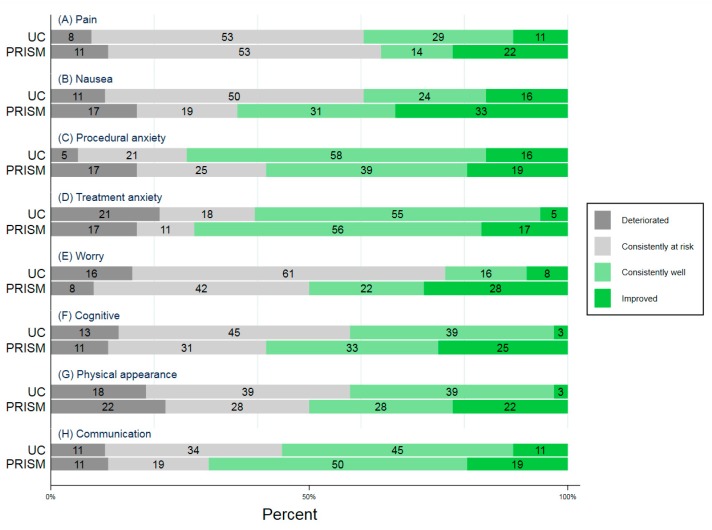
Cancer-specific HRQOL trajectories.

**Table 1 children-06-00124-t001:** Participant characteristics at time of enrollment, for patients who completed baseline and six-month assessments.

	At Baseline			At 6 Months		
**Characteristic**	Usual Care (*n* = 44)	PRISM (*n* = 48)	All (*n* = 92)	Usual Care (*n* = 38)	PRISM (*n* = 36)	All (*n* = 74)
	*n* (%)	*n* (%)	*n* (%)	*n* (%)	*n* (%)	*n* (%)
**Female**	24 (55)	16 (33)	40 (43)	21 (55)	12 (33)	33 (45)
**12–17 years-old at enrollment**	32 (73)	35 (73)	67 (73)	26 (68)	27 (75)	53 (72)
**18–25 years-old at enrollment**	12 (27)	13 (27)	25 (27)	12 (32)	9 (25)	21 (28)
**Non-White Race**	19 (43)	15 (31)	33 (36)	17 (45)	9 (25)	26 (35)
**Non-English First language**	10 (23)	1 (2)	11 (12)	9 (24)	3 (8)	12 (16)
**Leukemia/Lymphoma**	29 (66)	31 (65)	60 (65)	25 (66)	23 (66)	48 (66)
**Advanced Cancer at Enrollment**	14 (32)	10 (21)	24 (26)	11 (29)	6 (17)	17 (23)

**Table 2 children-06-00124-t002:** Percent positive trajectories (95% CI) for PedsQL SF-15 and Cancer Module subdomains, by study arm.

	UC		PRISM		
	Percent	95% CI	Percent	95% CI	P from Chi-Squared Test
**SF-15**					
**Global**	26	15–42	47	32–63	0.06
**Physical**	34	21–50	36	22–52	0.86
**Emotional**	37	23–53	58	42–73	0.06
**Social**	66	50–79	83	68–92	0.08
**School**	34	21–50	44	30–60	0.37
**Cancer Module**					
**Pain**	39	26–55	36	22–52	0.77
**Nausea**	39	26–55	64	48–78	0.04
**Procedural anxiety**	74	58–85	58	42–73	0.16
**Treatment anxiety**	61	45–74	72	56–84	0.29
**Worry**	24	13–39	50	34–66	0.02
**Cognitive**	42	28–58	58	42–73	0.16
**Physical appearance**	42	28–58	50	34–66	0.50
**Communication**	55	40–70	69	53–82	0.21

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
