# Peer review of "A Psychosocial Intervention’s Impact on Quality of Life in AYAs with Cancer: A Post Hoc Analysis from the Promoting Resilience in Stress Management (PRISM) Randomized Controlled Trial"

_children, 2019, doi:10.3390/children6110124_

Round 1

Reviewer 1 Report

This manuscript presents a post-hoc analysis from the PRISM trial, an intervention to build resilience in AYAs with cancer. In this manuscript the authors present effects of the PRISM intervention on Health-Related Quality of Life measures compared to usual care, descriptively comparing changes in scores across groups. This is a follow-up paper from the larger PRISM pilot RCT, for which the main findings and some secondary analyses have already been published. This is overall a very well written and strong manuscript. The introduction covers relevant previous findings, the study design is well described, and results are well reported. The reported results are in line with the pre-registered study outcomes on clinicaltrials.gov. I have only minor suggested changes to improve this already strong manuscript.

My most significant concern is the lack of attention paid to the cases in which the PRISM group displayed more negative trajectories than the usual care group. The authors briefly mention in the discussion section that more negative trajectories were observed for pain and procedural anxiety. The procedural anxiety findings are presented in the results section, but the pain findings are not mentioned at all in the text. The procedural anxiety results are also somewhat hidden amongst other, more positive findings. Furthermore, the authors suggest in the discussion section that these results may be because PRISM does not target these constructs, but in that case, would we not expect to see equivalent trajectories rather than more negative trajectories compared to usual care (which is an overall limitation of the fact that they only descriptively compare findings and cannot speak to statistically significant differences in these trajectories)? It may be that the PRISM intervention in fact confers harm for these domains, which needs careful monitoring in relation to adverse events during the larger RCT. Given the potential importance of these findings for the larger planned RCT, I recommend the following: Report the pain and procedural anxiety findings in a separate paragraph in the results section so that they are easily identified. Provide more discussion about how these findings are being considered with regards to study design and monitoring of harms for the larger RCT. The authors state that analyses were underpowered to detect differences in HRQoL, as the a priori analyses were powered based on detecting effects for the primary outcome (resilience). Why do the authors not perform post-hoc power analyses to indeed confirm that they were underpowered for secondary analyses, rather than just assuming that this is the case? Can the authors provide such post-hoc analyses to confirm lack of power? Can the authors provide references for any other published papers that use a similar descriptive analytic approach, rather than performing statistical comparisons, to situate their approach in the wider intervention literature? I think it would be helpful for the readers to better understand how the PRISM modules mapped onto the various HRQoL domains. It is clear from the manuscript that communication was a specific target, and thus it makes sense that this domain was improved in the PRISM group. However, are the authors able to provide more information about how the PRISM intervention did and did not target the other HRQoL domains? Were there specific expectations about which subdomains would improve? This information would be particularly helpful to interpret findings (i.e., were effects largest for those domains that were specifically targeted in the intervention), including to what extent the PRISM intervention seemed to improve domains that weren’t specifically targeted (eg., potential carry-over effects).

Author Response

Please see attached response.

Reviewer 2 Report

This article outlines post hoc analyses conducted to explore impact of Promoting Resilience in Stress Management (PRISM) intervention on AYA HRQoL. The authors are to be commended for their work in this under researched area. Overall, the paper is very well written. However, there are weaknesses within the current manuscript, which warrant consideration and are outlined below.

Main concerns:

The Table, Supplementary Tables, and Figure are not available for review. Without these data, it is difficult to provide a fully informed review of the manuscript.

Specific Comments:

Introduction: The introduction is clear and concise; however, the authors specify only 2 interventions aimed at improving psychosocial outcomes for AYAs with cancer and fail to mention several other relevant studies (for example: Robb SL, Burns DS, Stegenga KA, et al. Randomized clinical trial of therapeutic music video intervention for resilience outcomes in adolescents/young adults undergoing hematopoietic stem cell transplant: a report from the Children's Oncology Group. Cancer. 2014;120(6):909–917. doi:10.1002/cncr.28355). Further exploration of the literature in this area is warrented.

Methods: What were the internal consistencies for the total scores and subscale scores from the PedsQL General & Cancer module in this sample?

Discussion: Much of the discussion is a restatement of the results rather than a critical analysis of findings. As it is currently written, it does not clearly present a case for why these results are valuable (above and beyond previous analyses conducted to evaluate PRISM efficacy). The discussion would be strengthened by further articulation of what aspects of the PRISM intervention the authors suspect contribute to the positive trajectories on individual HRQoL subscales. Additional specific comment on how PRISM might be adapted/augmented based on these findings would also be helpful.

Author Response

Please see attached response.

Round 2

Reviewer 2 Report

Thank you for the opportunity to re-review. The authors have attending to the majority of reviewers’ concerns; however, they have not adequately addressed comments regarding request of internal consistencies for the total scores and subscale scores from the PedsQL General & Cancer module in this sample.

Given that the stated purpose of the manuscript is to describe changes in HRQOL subdomain scores, internal consistencies of the sample on the subdomains of interest are important missing data. As stated by Green et al (2006), “internal consistency is a property of the sample rather than the scale.” Authors are strongly encouraged to report and interpret these data in accordance with the “Summary of Practical Recommendations for Reporting, Analyzing, and Interpreting Reliability Data in Psychology Literature”.

Green, C. E., Chen, C. E., Helms, J. E., & Henze, K. T. (2011). Recent reliability reporting practices in Psychological Assessment: Recognizing the people behind the data. Psychological Assessment, 23(3), 656.

Author Response

We apologize for not addressing the reviewer’s concern in totality with our first revision. Thank you for the opportunity to make additional modifications. To address the reviewer’s concern, we have calculated the internal consistency of our sample for subscales and total scales on both instruments used. A sentence describing our analysis has been included in our methods and the results have been included accordingly. An additional supplementary table has been included as well, further detailing the complete results of this analysis. Thank you for the opportunity to carry out this additional analysis, we are further reassured that the instruments used appropriately reflect our intended measurements.
